# A Classification System for the Sustainable Management of Contaminated Sites Coupled with Risk Identification and Value Accounting

**DOI:** 10.3390/ijerph20021470

**Published:** 2023-01-13

**Authors:** Shiyi Yi, Xiaonuo Li, Weiping Chen

**Affiliations:** 1Laboratory of Soil Environmental Science and Technology, Research Center for Eco-Environmental Sciences, Chinese Academy of Sciences, Beijing 100085, China; 2College of Resource and Environment, University of Chinese Academy of Sciences, Beijing 100049, China

**Keywords:** contaminated sites, sustainable development, risk identification, classification management, cost–benefit analysis

## Abstract

Currently, site contamination is considered to be a sustained, international environmental challenge, and there is an urgent practical need to build a core theoretical system and technical methodology for the sustainable risk management of soil contamination, together with its prevention and control. We aim to improve the risk management of contaminated sites in the post-remediation era, in line with the current trend of sustainable development. The work is based on the theory of sustainability science and the eco-environmental zoning system., In this study, we build a conceptual model that can be used to classify the sustainable performance of contaminated sites in terms of risk management in line with the existing environmental management system for contaminated sites in China. To provide a scientific decision-making basis and technical support for the refined classification management of soil environments in China during the 14th Five-Year Plan period, five typical contaminated sites were selected for a quantitative evaluation by applying multi-technical approaches, including sociological, economic and statistical methods. The results showed that the sustainable performance of contaminated sites with regard to management was affected not only by pollution risk factors but also by potential utility benefits. Specified management strategies should be developed according to different levels of sustainability so as to achieve the goals of improving land use efficiency and enhancing urban functions.

## 1. Introduction

With the adjustment and transformation of industrial structures and the development of urbanization, many contaminated sites located on industrial land have caught stakeholders’ attention concerning the issues regarding their cleanup and redevelopment. Contaminated sites (also referred to as brownfield sites) are sites with pollution hazards beyond the acceptable risk levels for human or ecosystem health [1,2,3]. The implementation of contaminated site remediation has been practiced in developed countries for nearly half a century and evolved through a gradual change in management mindsets from the complete removal of contaminants to risk-based management and, finally, to green and sustainable risk management, supporting the wide-ranging consideration of risk management technology in the social, environmental and economic dimensions. As a result, in developed countries, organizations including ASTM, USEPA, ISO, CLARINET and SURFs have generated active responses by issuing a series of practice frameworks, standard guidelines and technical assessment guidelines to develop a systematic and comprehensive sustainable risk management system for contaminated sites [4,5,6,7,8]. The challenge is that the contradiction between the large number of sites and the shortage of resources, as well as the high cost of remediation and the urgent need for the elimination of high-risk sites worldwide, stakeholders (i.e., managers and experts), have made great efforts to explore risk-based contaminated site classification and management models to identify sites with high sustainable development potential. Such sites will then be afforded priority for remediation and redevelopment decisions to achieve the optimal allocation of limited resources and improve the effectiveness of risk communication and interactive decision-making among stakeholders.

Thus far, scholars have conducted a great deal of work on the risk classification of contaminated sites. From the perspective of the management of contaminated sites throughout the whole life cycle, risk classification is currently implemented through three basic measures: specific site zoning, based on health risk assessment; the classification of regional sites, based on risk identification; and classification management, based on the reuse of contaminated sites.

The first approach refers to specific site zoning based on health risk assessment. Since the contaminated site generally presents spatial heterogeneity in its contamination, it is necessary to identify contaminant characteristics (i.e., the types, distribution and boundary of the contamination) through an accurate site inventory to provide a scientific basis for refined health risk evaluation and subsequent zoning-based classification management [9]. Health risk assessment is an indispensable technical tool for developing a risk management system for contaminated sites; it contains four steps, comprising “hazard identification, dose–effect relationship, exposure assessment and risk characterization”, as proposed by the National Academy of Sciences (NAS) [10,11]. Based on the basic assessment procedure, a comprehensive risk management system for contaminated sites has gradually been established in Western countries, consisting of toxicity and physicochemical parameters, assessment methods and cleanup goals. Furthermore, the associated risk assessment models and software have been developed to facilitate practical applications, such as the RBCA (Risk-Based Corrective Action) model of the United States, the CLEA (Contaminated Land Exposure Assessment) model of the United Kingdom, the CSOIL (Contaminated Soil) model of the Netherlands and the HERA (Health and Environmental Risk Assessment Software for Contaminated Sites) model of China [12,13,14].

The other approach refers to the classification of regional sites based on risk identification. The U.S. was the first country to implement the risk-based management of contaminated sites by determining whether or not a site should be included in the National Priorities List (NPL) based on the comprehensive scoring mechanism of the Hazard Ranking System (HRS). As defined by the HRS, sites scoring higher than 28.5 should be listed in the NPL; otherwise, no further management actions will be taken [15]. Subsequently, Canada, the UK, France, Sweden and other countries also developed unique classification standards for contaminated sites and built corresponding database management systems for more effective site management decision-making. Among these, the HRS of the United States and the National Classification of Contaminated Sites (NCCME) of Canada are often employed for scientific research on contaminated site classification and management modes in other countries [16,17,18,19].

The third application of classification management is to enable the reuse of contaminated sites. Revitalizing the existing contaminated sites through risk management is of great practical importance for addressing the contradiction between land supply and demand and, thus, enhancing the efficient and rational use of land resources. However, not all contaminated sites are worthy of redevelopment. According to the “bathtub model” proposed by the Concerted Action on Brownfields and Economic Regeneration Network (CABERN), developers and other stakeholders lack the systematic consideration of the environmental conditions, social factors and economic benefits of site redevelopment, which can lead to less well-executed decision making regarding site risk management. Therefore, the major influential factors affecting site reuse planning, such as stakeholders’ attitudes, responsibility allocation, land policies and financial incentives, have been widely studied to realize the rationality of the decision-making outcomes [20,21,22,23,24,25,26,27,28,29]. For example, Bartke [30] used the Timbre brownfield prioritization tool to classify contaminated sites in the Czech Republic into five redevelopment priority levels, namely lower, low, medium, high and higher, in terms of the three dimensions of regional redevelopment potential, i.e., site attractiveness, market competitiveness and environmental risk elimination. Moreover, the spatial layout of infrastructures constraining land reuse planning, such as transportation, medical care and education, were taken into account to build a multi-objective decision-making model for optimal land planning [31,32,33,34,35,36,37].

Generally speaking, the sustainable management of contaminated sites is a complicated decision-making process. From a whole-life-cycle perspective, the early risk management strategies mainly focus on cost reduction and the minimization of adverse environmental impacts, while the post-remediation management strategies place greater emphasis on land benefits and social sustainability [38]. However, currently, studies on the risk classification of contaminated sites do not consider the interaction between potential risks and probable land values, which may lead to a biased decision-making process for site risk management, without considering the supporting roles of other decision dimensions (environment, society and economy). Therefore, to reveal the mechanism of interaction between risk management and regional sustainable development of contaminated sites, there is an urgent need to establish a decision-making support system for contaminated site management classification. With the aim of filling certain gaps in previous research and guide site sustainability management, this work is structured as follows: (1) The methodology used to develop an indicator system, process data, and quantify the evaluation indicators is presented; (2) Using the results, first, we conduct a detailed analysis of the risk characterization and land economic values of the selected contaminated sites, and then the category framework for the contaminated sites is mapped to explain the important factors that can significantly influence the sustainability levels and support of future site management decisions; (3) In the last section, the main conclusions are highlighted, and challenges to be addressed in future work are proposed.

## 2. Materials and Methods

### 2.1. Subsection

This study was conducted in five typical sites in China, i.e., the Xinguang Stainless Steel Factory (XSSF), the former Dongfang Chemical Plant (DCP), the Nanning Chemical Plant (NCP), Shenyang Chemical Plant (SCP) and the former Great Wall Chemical Plant (GWCP). The information collected on the sites included investigation and remediation reports, official government documents, yearbooks, questionnaire surveys and related websites. The location, basic information and risk management factors of the sites are shown in Figure 1 and Table 1.

### 2.2. Methods

#### 2.2.1. Assessment Indicator System 

The classification indicator system for contaminated site risk management was designed by combining sustainable development theory with a risk factor and cost–benefit analysis, including two groups: the risk index and value index. The risk index was used to evaluate the existing contamination levels, the vulnerability of the risk receptors and potential risks caused by operational enterprises., while the value index was applied to analyze the net benefit of site management in terms of the site remediation cost and the economic value of the land’s reuse. The definitions of the assessment indices are shown in Table 2.

#### 2.2.2. Data Collection and Accounting Methodology

Regarding the risk index, the calculation of the population density is based on the Resource and Environmental Science and Data Center (https://www.resdc.cn/, accessed on 10 August 2022). Using the Zonal Statistics as Table tool in ArcGIS, the raw data of the contamination level indicator are obtained directly from the site risk assessment report, and other risk indices are measured by collecting POI data from the Gaode open platform.

For the benefit index, the remediation cost can be extracted from the bidding documents of remediation projects, the land transfer price can be obtained from the Chinese land marketing website (https://www.landchina.com, accessed on 10 August 2022), and other benefit indices can be calculated with equations as follows below.

**Regional economic development**: A remediated site can not only eliminate health and ecological risks but also deliver social and economic benefits, such as the promotion of social stability and regional economic growth. In this study, the stigma effect of the contaminated site on the adjacent properties is predicted in order to evaluate the approximate economic development potentials that could be obtained through site management. The calculation of the stigma or rebound effect is as shown in the following equation:(1)$$EDV=D\mathrm{en}\times A_{1km}\times FS\times P\times p$$
where *Den* is the population density within 1 km of the studied site (person/km^2^), *A_1km_* is the land area within 1 km of the studied site (km^2^), *FS* is the floor space per capita (m^2^/person), *P* is the average house price within 1 km of the studied site (CNY/m^2^), and *p* is the house appreciation ratio (%).

**Job opportunity**: The dominant functions of the redevelopment of contaminated sites into commercial lands are the employment opportunities and the resulting increase in personal incomes. In this method, the market value approach is applied to evaluate the income increase by determining the average salaries of employees in various industries. Since this study focuses on urban commercial land, industries such as agriculture, forestry, animal husbandry and fisheries, mining, manufacturing, construction, transportation, storage and postal services are not included in the study boundary. The calculation of job opportunities is as shown in the following equation:(2)$$EV=\sum_{}^{}PL_{i}•SAL_{i}$$
where *PL_i_* is the average salary of industry *i* (10,000 CNY/person), and *SAL_i_* is the number of employees in the industry *i*.

**Recreation and leisure**: The recreation and leisure value represents the revenue generated by redeveloping contaminated sites into open green space, i.e., a park. Generally, such a value includes travel fees and consumer surplus, calculated indirectly by questionnaire surveys among park visitors. The calculation of the recreation and leisure value is as shown in the following equation:(3)$$V=\frac{(TC+CS)\times N}{n}$$
where *TC* is the travel cost (CNY), *CS* is the consumer surplus (CNY), *n* is the total number of questionnaire respondents, and *N* is the total number of park visitors in one year.

**The ecological value (including regulating services and supporting services)**: The change in land use can cause significant interactions between ecosystem services (ES) [39]. The major ES provided by contaminated site greening can be defined as regulating services (including gas regulation, climate regulation, environmental purification and hydrological regulation) and supporting services (including soil conservation, the maintenance of nutrient cycling and biodiversity) [40]. Based on land use/land cover (LULC), in this study, we estimated the ecosystem service values (ESV) based on the net primary productivity of the vegetation in the regenerated sites, using the method developed by Xie et al. [41]. The main LULC types involved in the ESV evaluation include mixed broadleaf–conifer forests, broad-leaved forests, shrubs, grasslands, scrubs, artificial lakes and wetlands. The calculation of the ESV is as shown in the following equation:(4)$$PESV=\sum_{n=1}^{7}AF_{\mathrm{ni}}D$$
where *A* is the area of the evaluated ecosystem (m^2^), *F_ni_* is the coefficient of the area values per unit of ES n in area *i*, and *D* is a standard equivalent factor of the ESV (CNY/m^2^). 

Considering the fact that the soil conservation service is affected by multi-complex factors and that soil conservation simulation data are almost entirely unavailable, the soil conservation service is not corrected in this study, while the area value per unit equivalent factors of other ES are adjusted by the NDVI (normalized difference vegetation index), or precipitation factor, in order to conform to the environmental situation of the studied site. The calculation of *F_ni_* is as shown in the following equation:(5)$$F_{ni}=\left\{\begin{array}{l} \frac{B_{i}}{B}\times F_{in1}\mathrm{or}\\ \frac{W_{i}}{W}\times F_{in2}\;\mathrm{or}\\ F_{in3} \end{array}\right.$$
where *B_i_* is the average NDVI of the evaluated ecosystem in area i in the last ten years, *B* is the average NDVI of the evaluated ecosystem in China in the last ten years, *W_i_* is the average precipitation in area *i* in the last ten years (mm·yr^−1^), *W* is the average precipitation in China in the last ten years (mm·yr^−1^), *F_in1_* represents ES including gas regulation, climate regulation, environmental purification, the maintenance of nutrient cycling and biodiversity, *F_in2_* represents hydrological regulation services, and *F_in3_* represents soil conservation services.

The net profit of crop production per unit area in the farmland ecosystem is used to measure a standard equivalent factor of the ESV [42]. The calculation of *D* is shown in the following equation:(6)$$D=\frac{1}{7}\sum_{i=1}^{n}\frac{m_{i}p_{i}q_{i}}{M}$$
where *m_i_* is the area of crop *i* (hm^2^), *p_i_* is the average price of crop *i* (CNY/ton), *q_i_* is the yield per unit area of crop *i* (ton/hm^2^), and *M* is the total area of all the crops (hm^2^).

**Normalization of the index**: Because the unit and numerical levels of an index are different, the raw data should be standardized to enable a comparative evaluation of different sites. The normalization of data is as the following equation:(7)$$\begin{array}{l} x^{*}=\frac{x-min}{max-min}(\mathrm{positive}\;\mathrm{indicator}),\\ x^{*}=\frac{max-x}{max-min}(\mathrm{negative}\;\mathrm{indicator}) \end{array}$$
where *x* is the initial value of the evaluation indicator, *max* represents the maximum value, and *min* represents the minimum value.

#### 2.2.3. Classification Method

Based on Equations (1)–(7), the standardized values of the risk and benefit indices could be weighted to obtain an overall risk value and benefit value ranging from 0 to 1. Then, the risk value was equally divided into three levels, i.e., low-risk (0–0.3), medium-risk (0.3–0.6) and high-risk (0.6–1). Similarly, the benefit value was classified into three levels, i.e., low-benefit (0–0.3), medium-benefit (0.3–0.6) and high-benefit (0.6–1) as shown in Figure 2. As a result, the sustainability performance of contaminated site risk management was categorized into three levels depending on both the risk value and benefit value, i.e., high-sustainability, medium-sustainability and low-sustainability, which are expected to provide support for scientific decision-making regarding future site managemen.

## 3. Results and Discussion

### 3.1. Risk Assessment of the Contaminated Sites

In terms of the vulnerability of risk receptors, potential contamination risk and current contamination level, the risk identification results of the five contaminated sites are shown in Table 3. The DCP, NCP and SCP sites are adjacent to the city center, meaning a greater population density (>1000), more sensitive objectives (>10) and more severe contamination (both soil and groundwater contamination) than those located in the suburbs, i.e., XSSF and GWCP, which demonstrates the higher risk posed by site contamination. Except for DCP, no contaminants are likely to be an input for the other four sites.

### 3.2. Benefit Assessment of the Contaminated Sites

The sustainability assessment of the economy is mainly based on the cost input and benefit output during the management of sites throughout their life cycles. In terms of economic loss, economic value, social value and ecological value, the value accounting results of the five contaminated sites are shown in Figure 3. 

The remediation actions and types of land reuse have important impacts on land values. As illustrated in Figure 3, although the investment for the remediation of the SCP site is significantly higher than that of the other sites, its redevelopment produces the highest direct land value, amounting to CNY 9552.91/m^2^. The assessment results show that the XSSF site has the highest social value, with CNY 8486.84/m^2^, due to its tourism and sightseeing functions, while the ecological value contributes least to the land value compared with the economic value and social value; NCP has the highest ecological value, with CNY 5.4/m^2^.

Based on face-to-face field interviews, the recreational benefits of the DCP were evaluated using Equation (3). The questionnaire survey was conducted on 2 April 2022 at the DCP site, and 140 questionnaires (out of a total of 150) were collected from the park visitors. No individual information was publicized in our study, and all the interviewees signed an “Informed Consent Form” before our investigation. Due to the limitation of the length of the paper, detailed information, including the questionnaire design, collected data and data analysis, are not provided here but can be accessed upon request to the authors. 

### 3.3. Benefit Assessment of the Contaminated Sites

Using Equation (7), a numerical value for a particular indicator was normalized against the maximum or minimum value to transform it into a score ranging from 0 to 1. As shown in Table 4, an indicator weight was assigned based on the hypothesis that each indicator belonging to the same criteria contributed equally to the site’s sustainability performance. For example, here, six risk indices contribute 1/6 (15~17%) to the total risk score.

The overall risk value, benefit value, rankings and classification of the individual sites are displayed in Figure 4. We can see that the classification can be equal between sites ranking differently in terms of their sustainability performance. As demonstrated below, the risk management actions of the XSSF, NCP and SCP sites are classified as high sustainability, while the sustainability performance of the DCP and GWCP is relatively low. It is clear that the classification results depend on both the risk characteristics and the benefit feedback.

The risk value assessment results in Figure 4 show that the contamination risks of the NCP and SCP sites are higher than those of the other three sites due to the fact that sensitive objectives are crowded around the two chemical plants, especially primary and secondary schools, which number as high as 55 and 43, respectively. Therefore, the risk receptors around the NCP site and SCP site show high vulnerability, and the areas impacted by contamination risks are enlarged due to local overpopulation. Comparatively, the small population and number of sensitive objectives around XSSF mean that it has the lowest risk value among all the sites. In addition, the unacceptable human health risk posed by the groundwater in the SCP, NCP and DCP sites results in significant differences in the risk value, indicating that contaminants in the groundwater tend to contribute more than those in the soil to risk formation and transmission.

In terms of the benefit value assessment results in Figure 4, the net benefits of the remediation actions of the SCP, XSSF and NCP sites are significantly higher than those of DCP and GWCP. Specifically, the redevelopment of the SCP site into residential land generates immense appreciation effects on the remediated site and surrounding properties (CNY 9073.29/m^2^), though this also results in the highest remediation cost per unit area (CNY 480.33/m^2^). In the case of XSSF, the job opportunities and tourism revenue form a greater proportion of the total value due to the site’s particular location in a tourism city (Huangshan City). Meanwhile, for the NCP site, located in the city center, the high land transfer price and low remediation cost (CNY 35.13/m^2^) enable a relatively higher net benefit (CNY 7094.55/m^2^). Currently, there is no reuse plan for the GWCP site; however, considering its geographical location and risk management countermeasures, an industrial purpose may be its ideal reuse scenario and could produce social benefits by providing job opportunities. Although the DCP site has great property appreciation (CNY 5.39 billion), the park reuse scenario limits its market value and economic output, thus resulting in the lowest net benefit (CNY 768.38/m^2^).

## 4. Conclusions

This study established a value-based approach to improve the existing risk classification system for sustainable site regeneration, through which contaminated sites with high pollution risks and high reuse values can be identified in order to implement risk control activities and subsequent development planning. In particular, for regions with numerous contaminated sites subjected to financial challenges and land shortages, the novel classification system has a wider range of applications; not only could it eliminate the pollution risks, but it also could maximize the land utilization profits. The main conclusions regarding the case study of the five contaminated sites are as follows:(1)The size of the population and the number of sensitive objectives have a major impact on the contamination risk levels. At present, the human health risk is the primary concern of the risk assessment of contaminated sites. Contaminated media can lead to significant risk differences; groundwater pollution has greater effects in changing the risk values compared with soil pollution.(2)The overall value of a site depends on its economic value, social value and ecological value. Among these factors, the appreciation potential of the site and surrounding properties as a result of site remediation actions plays a vital role in increasing the overall value of the site. The value produced by providing job opportunities is an important aspect of the social value, and the ecosystem service value accounts for a relatively small proportion of the total value.(3)The sites should be treated differently according to the corresponding risk and benefit evaluation results. For sites with high sustainability, decisions regarding their prior risk management and reuse should be made as soon as possible to eliminate health risks and generate substantial profitability through land reuse. For sites with a medium level of sustainability, a restricted development strategy is recommended, and management patterns with higher returns should be actively explored for the purpose of eliminating existing risks. For sites with low sustainability, management decisions should focus on preventing contamination diffusion rather than rapid redevelopment while simultaneously conducting a scenario analysis to maximize the land values.

Compared with previous studies, our research differs in two aspects: Firstly, the risk evaluation depended on indicators measuring the existing risks and the potential risks posed by local pollution sources, which enabled a dynamic simulation according to the spatial emission characteristics of the contaminants. Secondly, by focusing on the current challenges involved in sustainability assessments for brownfield regeneration, this paper proposed a quantitative method that can be used to monetize the social, economic and environmental benefits of site management, which performs better than subjective approaches based on expert scoring or descriptive arguments. However, several challenges remain to be addressed in further research. For example, the evaluation pattern is strongly dependent on the indicators used, but here, further analysis of the possibly more influential indicators was not discussed in depth due to the scarcity of data. Additionally, the weights of both the overall index and sub-indices are hypothesized to be equal according to the calculation process, although they actually have different contributions to sustainability performance. Moreover, the indicator of the potential risks is measured by the input flux of contaminants rather than the number of surrounding enterprises.

## Figures and Tables

**Figure 1 ijerph-20-01470-f001:**
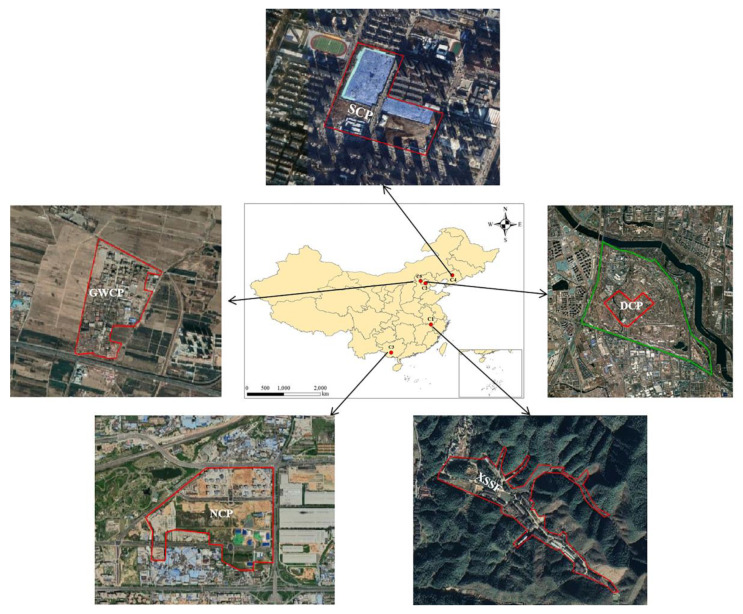
Geographic locations of the studied sites.

**Figure 2 ijerph-20-01470-f002:**
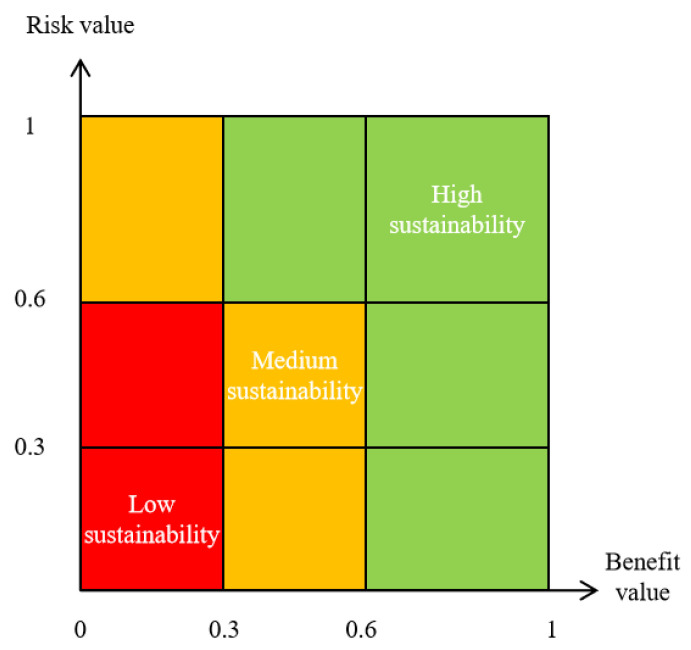
The classification interval for contaminated site risk management.

**Figure 3 ijerph-20-01470-f003:**
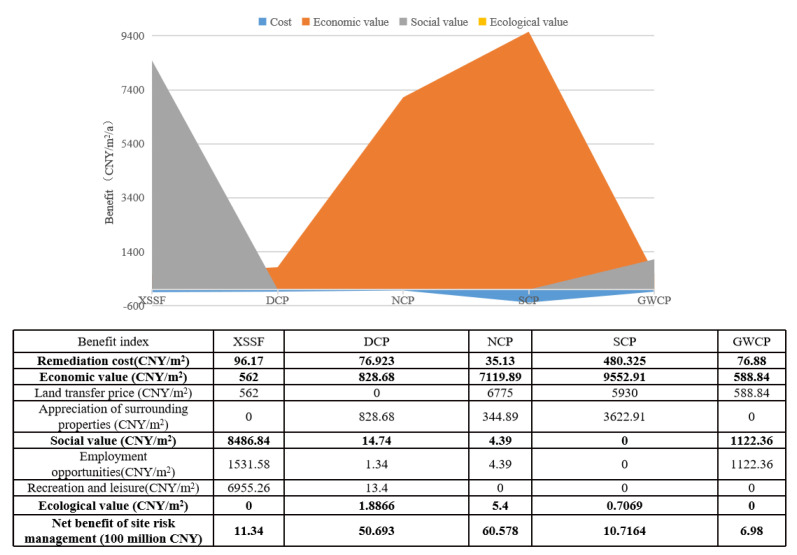
The results of the benefit assessment of the contaminated sites.

**Figure 4 ijerph-20-01470-f004:**
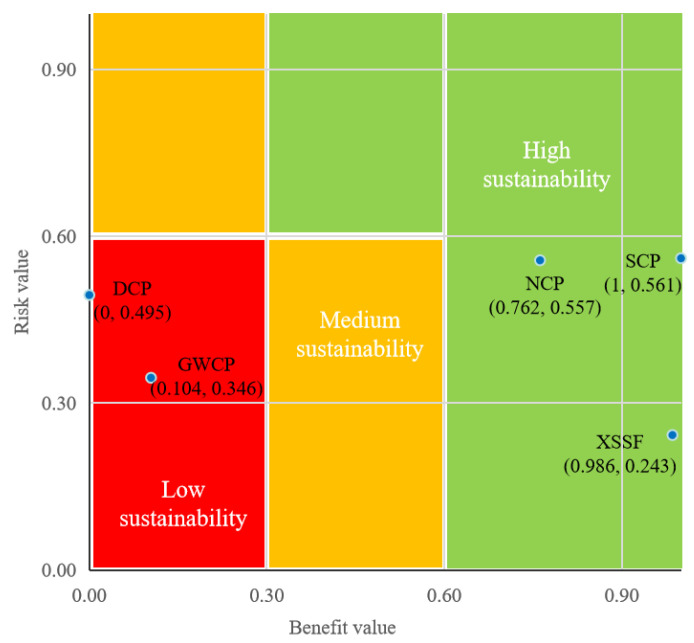
Sustainability performance of the contaminated sites’ risk management.

**Table 1 ijerph-20-01470-t001:** Basic information of the studied sites.

Site	Area (10,000 m^2^)	Contaminants	Treatment Technology	Cost (10,000 CNY)	Land Planning
XSSF	12.67	Soil: Ni, Cr^6+^, benzo (a) pyrene, benzo (a) anthracene, etc.	Solidification/stabilization, barrier and landfill	1218.19	Commercial and service facilities
DCP	650	Soil: benzene; Groundwater: benzene	Horizontal barrier, institutional control, ecological restoration, natural attenuation, long-term monitoring	50,000	Park and green space
NCP	85.387	Soil: heavy metals, SVOCs, VOCs and petroleum hydrocarbons (<C16); Groundwater: vinyl chloride, carbon tetrachloride, etc.	Solidification/stabilization, chemical oxidation, thermal desorption	3000	Park and green space, commercial, residential and service facilities
SCP	18.84	Soil: heavy metals, VOCs; Groundwater: pentachlorophenol, etc.	Solidification/stabilization, chemical oxidation, landfill	9049.32	Residential land
GWCP	42.71	Soil: VOCs, SVOCs, organochlorine pesticides and dioxins	Horizontal barrier, ecological restoration, long-term monitoring	3283.49	Construction land

**Table 2 ijerph-20-01470-t002:** Sustainable risk management zoning indicators for contaminated sites.

The 1st-Level Index	The 2nd-Level Index	The 3rd-Level Index	Definition
Risk index	Vulnerability of risk receptors	Population density	Population located within 1 km of the site.
		Distribution of sensitive objectives	The number of land use scenarios within 1 km of the site: primary and secondary schools, medical and health care and social welfare facilities.
		Ecological function areas	The number of ecological function areas (listed in spatial planning reports) within 1 km of the site: water conservation, biodiversity protection, flood regulation and the supply of agricultural products.
	Potential contamination risk	Number of operational enterprises	The number of operational enterprises within 1 km of the site: non-ferrous metal mining and processing, petroleum processing, chemical, coking, electroplating, etc., or those that have caused environmental pollution in recent years.
	Contamination level	Soil contamination	Number of contaminants in soil exceeding the relative standards, as recorded in remediation reports.
		Groundwater contamination	Number of contaminants in groundwater exceeding the relative standards, as recorded in remediation reports.
Benefit index	Economic loss	Remediation cost	The cost required to remove existing contaminants.
	Economic value	Regional economic development	The edge effects that could be produced through surrounding environmental improvements, i.e., house appreciation.
		Land price	The transfer price of the land after remediation.
	Social value	Employment opportunities	The value of the job opportunities that could be provided by redeveloping a remediated site for a commercial purpose or into a park.
		Recreation and leisure	The value of the recreation and leisure services that could be provided by redeveloping a remediated site into a park.
	Ecological value	Regulating services	Mainly includes gas regulation, climate regulation, environmental purification and hydrological regulation.
		Supporting services	Mainly includes soil conservation, the maintenance of nutrient cycling and biodiversity.

**Table 3 ijerph-20-01470-t003:** The results of the risk identification analysis of the contaminated sites.

Risk Index	XSSF	DCP	NCP	SCP	GWCP
**Population density within 1 km of the site (person/km^2^)**	127	2336	1195	3427	239
**Number of sensitive objectives within 1 km of the site**	0	13	56	53	0
Primary and secondary schools	0	8	55	43	0
Medical and health sites	0	2	1	5	0
Social welfare sites	0	3	0	5	0
**Number of ecological function areas within 1 km of the site**	1	1	0	0	1
Water conservation	1	0	0	0	1
Biodiversity protection	0	0	0	0	0
Flood regulation	0	1	0	0	0
Supply of agricultural products	0	0	0	0	0
**Number of operational enterprises within 1 km of the site**	0	8	0	0	0
**Soil contamination of the site**	10	1	21	11	22
**Groundwater contamination of the site**	0	1	8	7	0

**Table 4 ijerph-20-01470-t004:** The standardized data and weights of the evaluation index.

Indicator	Weight	XSSF	DCP	NCP	SCP	GWCP
Population density	0.17	0.000	0.669	0.324	1.000	0.034
Number of sensitive objectives	0.17	0.000	0.232	1.000	0.946	0.000
Number of ecological function areas	0.17	1.000	1.000	0.000	0.000	1.000
Number of operational enterprises	0.15	0.000	1.000	0.000	0.000	0.000
Number of soil contaminants	0.17	0.429	0.000	0.952	0.476	1.000
Number of groundwater contaminants	0.17	0.000	0.125	1.000	0.875	0.000
Net benefit	1	0.9855	0.0000	0.7617	1.0000	0.1043

## Data Availability

Even though the data is not publicly available it can be requested to the corresponding author.

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
