# Peer review of "A Classification System for the Sustainable Management of Contaminated Sites Coupled with Risk Identification and Value Accounting"

_ijerph, 2023, doi:10.3390/ijerph20021470_

Round 1
Reviewer 1 Report
Please edit this paper for english prose, grammar and spelling. Additionally, please write a proper discussion comparing the results to the literature, as of now you have only reported the results.
This paper is not publishable in its current form.
Reviewer 2 Report
The work deals with a very interesting problem related to determining the direction of development of polluted areas. Sustainable management of contaminated sites is proposed to be combined with risk identification and, at the same time, value accounting.
This seems to be a very desirable approach nowadays.
The work was well written, in a clear and legible way. The introduction of the work was correctly outlined. The methods are presented in a clear and exhaustive way. The only downside may be the small amount of data, which the authors point out.
Reviewer 3 Report
interesting study and good to focus on sustainability of remediation. however as with all models, there are limitations and this does not seem to be evaluated. the approach is not a very critical analysis and evaluation
Table 2: you mention comtamination as an indicator. these are remediated sites - so how do you define this aspect - is it from the reports from the remediation treatments applied? how are ecological indicators/value defined - there are ambiguous statements about sensitivity and values but no definition of what they specifcally are. more explanation and specific site relevant details
line 38 on - references to previous studies providing guidelines is not clear - references incomplete and incorrect, review documentation e.g. https://doi.org/10.1002/rem.21587 is a better place to start.
line 174 - recreation and leisure - mentions questionnaire. No details of survey provided. ethical approval?
conclusions - very vague and non specific. what is the value of the study ? how is it applied, what data needed to reinfe the analysis.
Round 2
Reviewer 1 Report
I have responded to your replies as stated in the word document:
1) The editing of this manuscript still needs improvement. It is not a question of American to British spelling; I use the latter. The opening paragraph alone has several incorrect uses of words and grammar or prose errors. There continue to be errors in the other parts of the manuscript; if you used a paid editing service, have them re-edit your paper. It is not relevant that other reviewers indicated that the article was "well-written," given the errors that start immediately within the first few lines of the manuscript.
2) A discussion is a crucial part of any article; you should convey the importance of your research here. In my many years in risk assessment, I have been involved in numerous high-level studies published in top-tier journals that are cutting edge. Your argument, "results are unable to be compared with other literature," is incorrect. Your discussion should focus on how your results differ from the existing literature, methods or techniques. Try to tease out exactly what makes your research unique from what already exists; as of now, you present results, and I cannot elucidate the point of your study. The significance of this research remains unclear.
Interpret your results (what do they mean in the bigger picture?), explain the implications of your findings (where and how do they apply?), and what are the future directions of this research (as of now, all I can see is you need more indicators and samples...that's uninteresting)?
In this case, my first decision stands, and I believe this manuscript is not publishable in its current form.
Author Response
1)The editing of this manuscript still needs improvement. It is not a question of American to British spelling; I use the latter. The opening paragraph alone has several incorrect uses of words and grammar or prose errors. There continue to be errors in the other parts of the manuscript; if you used a paid editing service, have them re-edit your paper. It is not relevant that other reviewers indicated that the article was "well-written," given the errors that start immediately within the first few lines of the manuscript.
Response: thanks for your suggestion, and we are sorry for the numerous errors, following the comment, we have used the editing service of MDPI to re-edit the paper, and the certificate has been uploaded with the revised submission.
2) A discussion is a crucial part of any article; you should convey the importance of your research here. In my many years in risk assessment, I have been involved in numerous high-level studies published in top-tier journals that are cutting edge. Your argument, "results are unable to be compared with other literature," is incorrect. Your discussion should focus on how your results differ from the existing literature, methods or techniques. Try to tease out exactly what makes your research unique from what already exists; as of now, you present results, and I cannot elucidate the point of your study. The significance of this research remains unclear.
Interpret your results (what do they mean in the bigger picture?), explain the implications of your findings (where and how do they apply?), and what are the future directions of this research (as of now, all I can see is you need more indicators and samples...that's uninteresting)?
Response: thanks for your suggestion, and yes, the current conclusion of this manuscript is a simple description of the case studies rather than an in-depth extension, therefore, following the comment, we revised the Conclusion to emphasize the significance of our study, application of the method, limitations need to be improved in future work, and superiority compared with the existing studies.
